# Unscented Kalman Filter Based on Spectrum Sensing in a Cognitive Radio Network Using an Adaptive Fuzzy System

**Md Ruhul Amin [1,‡], Md Mahbubur Rahman [1,‡], Mohammad Amazad Hossain [2,†,‡], Md Khairul Islam [3,‡], Kazi Mowdud Ahmed [1,‡], Bikash Chandra Singh [1,4,‡] and Md Sipon Miah [1,*,†,‡]**

[1] Department of Information and Communication Engineering, Islamic University, Kushtia 7003, Bangladesh; ruhulamin.ice@gmail.com (M.R.A.); mrahman@ice.iu.ac.bd (M.M.R.); mowdud@ice.iu.ac.bd (K.M.A.); bcssingh@uninsubria.it (B.C.S.)

[2] Department of Information and Communication Engineering, Noakhali Science & Technology University, Sonapur 3814, Noakhali, Bangladesh; m.hossain3@nuigalway.ie

[3] Department of Biomedical Engineering, Islamic University, Kushtia 7003, Banglasesh; khairul@bme.iu.ac.bd

[4] DiSTA, University of Insubriaz, 21100 Varese, Italy

[*] Correspondence: m.miah1@nuigalway.ie; Tel.: +353-838607424

[†] Current address: Discipline of Information Technology, National University of Ireland Galway, H91 CF50 Galway, Ireland.

[‡] These authors contributed equally to this work.

**Abstract:** In this paper, we proposed the unscented Kalman filter (UKF) based on cooperative spectrum sensing (CSS) scheme in a cognitive radio network (CRN) using an adaptive fuzzy system—in this proposed scheme, firstly, the UKF to apply the nonlinear system which is used to minimize the mean square estimation error; secondly, an adaptive fuzzy logic rule based on an inference engine to estimate the local decisions to detect a licensed primary user (PU) that is applied at the fusion center (FC). After that, the FC makes a global decision by using a defuzzification procedure based on a proposed algorithm. Simulation results show that the proposed scheme achieved better detection gain than the conventional schemes like an equal gain combining (EGC) based soft fusion rule and a Kalman filter (KL) based soft fusion rule under any conditions. Moreover, the proposed scheme achieved the lowest global probability of error compared to both the conventional EGC and KF schemes.

**Keywords:** cognitive radio network; spectrum sensing; Kalman filter; extended Kalman filter; unscented Kalman filter; fuzzy system

---

## 1. Introduction

Wireless sensor networks (WSNs) consist of spatially distributed self-governing sensors [1,2], i.e., sensors which monitor physical or environmental conditions. One of the big challenges is the deficiency of the spectrum available for wireless devices in a WSN.

Nowadays, the federal communication commission (FCC) has reticent 900 MHz licensed spectrum bandwidth at a center frequency of 2.4 GHz (i.e., IEEE 802.15.4 standards) which allows short-range communication in a WSN [3]. However, the demand of radio frequencies has tremendously increased due to different kinds of wireless applications not being available for fulfilling the users demand in terms of wireless communication perspectives. Moreover, the increasing number of wireless applications will also give channel dissension and spectrum deficiency. As a result,

there is a sharp increase in the demand for spectrum resources, which satisfies their wireless communication prerequisites.

To mitigate the aforementioned problem, the cognitive radio (CR) is a new technology that improves the spectrum efficiency for a WSN. In a cognitive radio network (CRN), a CR-enabled sensor in a WSN is called a cognitive user (CU) in which each CU opportunistically accesses the unused licensed spectrum if the licensed spectrum is vacant by the primary user (PU) [4,5]. Consequently, the spectrum sensing scheme is the key component of a CRN. It plays a vital role in detecting the spectrum band in order to allow the CU to access the unoccupied spectrum band without harming licensed PUs [6]. On the other hand, when the PU comes back into operation, the CU should instantly vacate the licensed spectrum due to avoiding interference with the PU [7]. To avoid interference with the PU, the spectrum sensing scheme must be more effective and efficient for a CRN. In a CRN, the spectrum sensing techniques are classified: cyclostationary detection, matched filter detection, Eigenvalue based detection and energy detection (ED).

In cyclostationary detection, it offers good detection performance [8] when it knows the information of the PU cyclic frequencies. However, it needs a longer time slot for the sensing phase in frame structure. In matched filter detection, it achieves a desirable value of the detection gain compared to other techniques during a short sensing time [9]. However, it needs the complete information of the PU's signaling features. In Eigenvalue based detection, it is one of the most recent and promising techniques [10,11], where the test statistic is computed from the eigenvalues of the received signal sample covariance matrix. Moreover, it does not need prior information of sampling signals. In ED, it is an attractive and simple technique because it does not need any prior information of the licensed signal; it is easy to accomplish with modest complexity [12]. However, the detection gain is compromised under the different channels i.e., fading, shadowing, uncertainty and the hidden terminal problem.

To mitigate this drawback, a cooperative spectrum sensing (CSS) scheme is analysis in [13] which enhances the sensing gain by using the spatial diversity where CUs are distributed located. Like a CSS scheme, each CU performs sensing to determine the status of the PU locally and transmits their received sensing measurements to the corresponding fusion center (FC), which combines them and makes a global decision based on fusion rules [14,15] that are classified into two categories i.e., soft fusion rule and hard fusion rule. In a soft fusion rule, it gives better detection gain compared with the hard fusion rule. In addition, it decreases the probability of error as compared to the hard fusion rule. However, this fusion rule requires larger overhead to transmit the accurate sensed energies to the FC compared to the hard fusion rule. In a hard fusion rule, it requires smaller overhead to share the local decision to the FC compared to the soft fusion rule. However, it degrades the detection gain compared with the soft fusion rule. In addition, it increases the probability of error as compared to the soft fusion rule. In a noise uncertainty environment, the detection gain in a CSS scheme is very poor [16,17] due to each CU obtaining low signal-to-noise ratio (SNR) information of the license PU signal, i.e., $-28$ dB [14].

The challenging job is to reduce SNR wall with a given detection probability and therefore is robust to noise power uncertainty [18,19]. Another challenge is to estimate uncertainty for both linear and nonlinear systems [20]. A fuzzy fusion was proposed to make a local soft decision at a CU by considering the SNR of the PU signal that is familiar to a CU [21]. Most existing CSS schemes were based on the assumption that the SNR of the PU at the CU was fully familiar [22]. However, it is very difficult to calculate the SNR value of the PU signal in a given spectrum band as there is no cooperation between the CU and PU. However, the CUs can calculate these parameters very well; it is very difficult to transmit their local measurements to the corresponding FC through the dedicated control channel.

To mitigate the impending spectrum starvation problem, an adaptive CSS scheme is proposed to detect spectrum channels accurately under the conditions i.e., the prior information, the prior activity and known SNRs of the PU signal. If these conditions are not available at CUs [23]. Sometimes, the detection gain degrades under different fading channels and becomes difficult to estimate the true state of the system [24]. In a CRN, each CU calculates the energy of its received signal and then forwards

their local measurements to the corresponding FC. Data fusion at the FC is performed by an adaptive fuzzy system where fuzzification parameters are adapted from received data via Unscented Kalman filter (UKF).

The KF is a popular method that is widely used for tracking and estimation due to its simplicity, optimality, tractability and robustness [25,26]. However, the application of the KF in a nonlinear system can be difficult. On the other hand, the extended Kalman filter (EKF) is derived from the conventional KF that simply linearizes all nonlinear models [26,27]. However, it is only applicable for reliable linear system using the time scale of the update intervals. In addition, it is difficult to implement and tune. Therefore, both the KF and EKF do not work in a nonlinear system due to both failing to converge to the true value.

For these reasons, we proposed the UKF based on spectrum sensing in a CRN using an adaptive fuzzy system in a nonlinear system, where the unscented transformation is a promising method for computing the value of a random variable in a nonlinear system that undergoes a nonlinear transformation.

The major contributions of our paper are as follows:

The proposed UKF scheme for a nonlinear system is to minimize mean square estimation error.

We proposed UKF scheme based on CSS in a CRN using an adaptive fuzzy system where, firstly, we developed Algorithm 1 and, secondly, the fuzzification parameters are adapted based on Algorithm 1 from received data at the FC.

---

**Algorithm 1** In the proposed scheme based on the UKF, all CUs are calculated the UKF gain and the estimated covariance.

---

**Input: Select appropriate segma points,** $UKF\left(x_0^i, p_0^i\right)$

**Output: Calculate the UKF gain,** $K$ **and the estimated covariance,** $p(k+1)$

1: Initialize:

$$x^-(k-1)=\begin{bmatrix} x_0^i & 0 \end{bmatrix}; i = 1, 2, ..., M$$

$$p(k-1) = \begin{bmatrix} p_0^i & 0 \\ 0 & p_v \end{bmatrix}$$

2: Loop:

3: Selecting sigma points:

$$x\left(j,k\right)=x^-\left(k-1\right)+\left(\sqrt{(n+s)\,p\,(k-1)}\right)_{(j)}; j = 1, 2, ..., n$$

$$x\left(j+n,k\right)=x^-\left(k-1\right)-\left(\sqrt{(n+s)\,p\,(k-1)}\right)_{(j)}; j = n+1, n+2, ..., 2n$$

4: Prediction step:

$$x\left(j,k+1\right)=F\left(x\left(j,k\right)\right)$$

$$x^-\left(k\right)=\sum_{j=1}^{2n} w_j^m x\left(j,k+1\right)$$

$$p^-(k)=\sum_{j=1}^{2n} w_j^m (x(j,k+1) - x^-(k))(x(j,k+1) - x^-(k))$$

5: Updating step:

$$y\left(j,k+1\right)=H\left(x\left(j,k+1\right)\right)$$

$$y^-\left(k\right)=\sum_{j=1}^{2n} w_j^m y(j,k+1)$$

$$p_y=\sum_{j=1}^{2n} w_j^c (y(j,k+1) - y^-(k))(y(j,k+1) - y^-(k))$$

$$p_{xy}=\sum_{j=1}^{2n} w_j^c (x(j,k+1) - x^-(k))(y(j,k+1) - y^-(k))$$

$$K=\frac{p_{xy}}{p_y+R}$$

$$x^-\left(k+1\right)=x^-\left(k\right)+K\left(y\left(j,k\right) - y^-\left(k\right)\right)$$

$$p\left(k+1\right)=p^-\left(k\right)-Kp_yK^T$$

6: Go to step-2

7: End

---

For detection gain, simulation results show that our proposed scheme have obtained better detection gain compared with both the EGC based soft fusion rule and KL based soft fusion rule under different channel conditions i.e., additive white Gaussian noise (AWGN) fading channel, Rayleigh fading channel, and Log-normal shadow fading channel.

Based on the detection gain, we can analyze global probability of error of the proposed UKF scheme which obtained the lowest compared with both the EGC and KL based soft fusion rule.

The rest of this paper is organized as follows: the general motivation and the background of this paper is outlined in Section 2. Section 3 gives a brief our system model. In Section 5, we proposed unscented Kalman filter (UKF) using adaptive fuzzy systems. Then, Section 6 provides simulation results and the performance of the proposed scheme. Finally, Section 7 concludes the paper and gives some directions for future research.

In addition, all parameters used in this manuscript are listed in Table 1 as follows.

**Table 1.** Main parameters.

| Parameters | Meaning |
|---|---|
| $K$ | The Kalman gain |
| $w^0$ | The effective weights |
| $H(H_0/H_1)$ | Hypotheses (absent/present) |
| $p_E$ | The global error probability |
| $\tau_{th}$ | The global decision threshold at the fusion center (FC) |
| $x^-(k)$ | The posterior estimate at the $k$th element |
| $p^-(k)$ | The posterior covariance at the $k$th element |
| $y^-(k)$ | The posterior observations at the $k$th element |
| $F(.)$ | The nonlinearity function in the process model |
| $H(.)$ | The nonlinearity function in the measurement model |
| $\lambda_i$ | The signal-to-noise ratio (SNR) at the $i$th cognitive user (CU) |
| $pd_i$ | The local decision based on the observation at the $i$th CU |
| $gd_f$ | The global decision at the FC where the subscript, $f$ is the probability of false alarm |
| $gd_d$ | The global decision at the FC where the subscript, $d$ is the probability of detection |

## 2. Related Work

The authors conducted an analysis of non-CSS in a CRN [28,29]. However, when the spectrum sensing is applied to a single CU under the time-varying channel i.e., fading, shadowing, where each CU can not distinguish the spectrum hole under a deeply faded and shadowed PU signal. In [30,31], the authors proposed a cooperative scheme based on Eigenvalue-based spectrum sensing (EBSS) techniques, which are applied in a completely decentralized manner. To overcome this problem, the authors conducted an analysis of CSS in a CRN [32]. In a CSS scheme [33–36], the detection gain is compromised under different channels i.e., fading, shadowing, uncertainty and the hidden terminal problem. In this paper, the authors investigate the issue of efficient and robust cooperative spectrum sensing in dense cognitive vehicular networks [37]. In this paper, the authors derive the optimal values for both the transmission time and the sensing time in presence of PU mobility [38]. A comparison between binary and continuous genetic algorithms for cooperative spectrum optimization in a CRN is presented in [39]. In [40], the authors are analytically derived with the object of maximizing the detection accuracy in the realistic environment. The authors conducted an analysis of an adaptive threshold and optimal frame duration for multi-taper spectrum sensing in CR [41]. Dynamic threshold based CSS using a coalitions game for CRNs is presented [42]. The authors are investigated a priori to determine on-demand access. In addition, it performs by accounting for the correlation exhibited by primary traffic patterns [43]. The authors [44] present the fuzzy logic based spectrum handover approach in CRN: A Survey. The authors [45] presented a CSS scheme based on trust and fuzzy logic for CRNs. A fuzzy logic based decision system for context aware cognitive waveform generation is presented in [46]. A fuzzy inference system was proposed in [47,48], which makes a local soft decision

at CU under the assumption whenever the SNR of the PU signal is familiar to CU. However, the convergent speed of the estimation algorithm is higher.

## 3. System Model

The proposed system model consists of a primary network (PN) and a CRN as shown in Figure 1 where each CU opportunistically accesses the licensed spectrum of the PU. The PN consists of PUs such as primary transmitters and receivers. The operation of the PU and the CU are executed based on the time division multiplexing access (TDMA). In a CRN, it consists of $M$ collaborative CUs such as including CU transmitters and CU receivers.

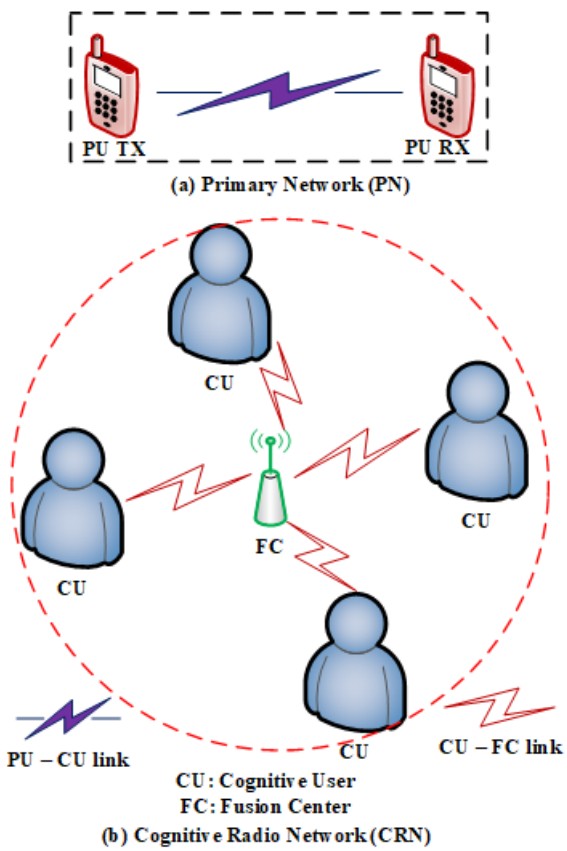

**Figure 1.** The proposed system model.

Under the frame structured as shown in Figure 2, each CU senses the PU channel during the sensing time slot. In the reporting time slot, each CU transmits their sensing results of the received PU signal to an FC which combines the received signals and makes a global decision and then broadcasts it back to all CUs.

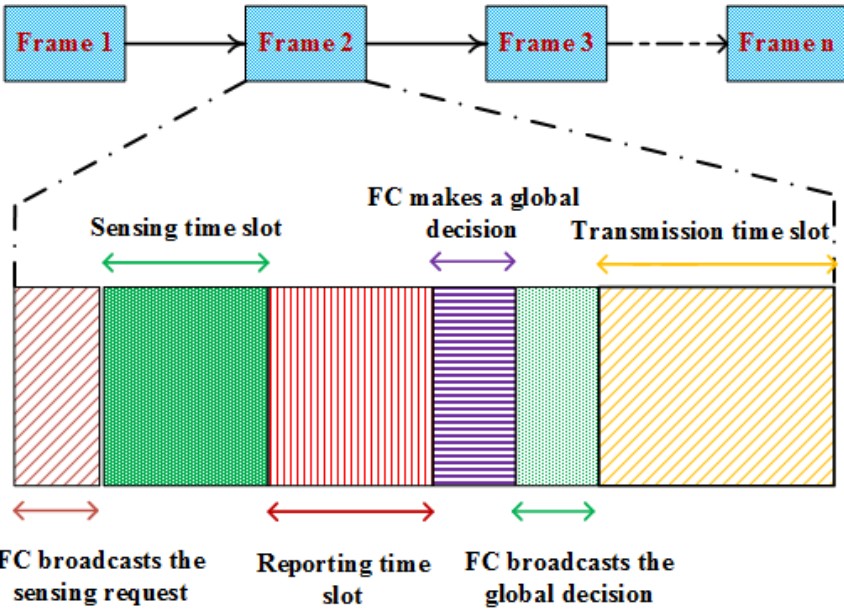

**Figure 2.** The frame structure of the proposed scheme.

Let $H_0$ and $H_1$ be the hypotheses that represent the absence and presence of the PU signal, respectively. The spectrum sensing can be formulated under this binary hypothesis as follows [14,15]:

$$\begin{cases} H_0: & \text{if PU is absent,} \\ H_1: & \text{if PU is present.} \end{cases} \tag{1}$$

Based on the packet transmission of the PU, the sensing results $r_i(t)$ at the $i$th CU can be defined under this binary hypothesis as follows:

$$r_i(t) = \begin{cases} w_i(t); & : H_0, \\ s(t)h_i(t) + w_i(t); & : H_1, \end{cases} \tag{2}$$

where $r_i(t)$ represents the sensing results at the $i$th CU, $s(t)$ represents the PU transmitted signal which is a binary phase-shift keying (BPSK) signal, $h_i(t)$ represents the channel gain between the $i$th CU and PU, and $w_i(t)$ represents the channel noise at the $i$th CU.

## 4. Energy Detection Technique

In this section, we consider that each CU transmitter senses the PU signal of the proposed scheme using the ED technique [14,15]. The general block diagram of the ED technique is shown in Figure 3 where the sensing result, $r_i(t)$ received at the $i$th CU transmitter, a band pass filter is applied to the received signal, and then the output of this filter is transformed by an analog-to-digital converter (ADC) which will be individually averaged and squared to estimate its own measured energy as follows:

$$\begin{aligned} r_1 &= |r_1(0)|^2 + |r_1(1)|^2 + |r_1(2)|^2 + ... + |r_1(N-1)|^2, \\ r_2 &= |r_2(0)|^2 + |r_2(1)|^2 + |r_2(2)|^2 + ... + |r_2(N-1)|^2, \\ r_M &= |r_M(0)|^2 + |r_M(1)|^2 + |r_M(2)|^2 + ... + |r_M(N-1)|^2. \end{aligned} \tag{3}$$

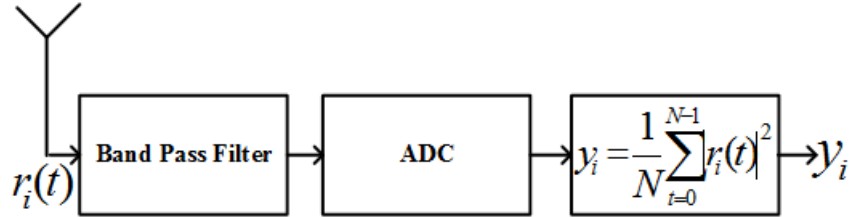

**Figure 3.** The energy detection technique.

From Equation (3), we can calculate the received signal, $y_i$ at the $i$th CU as follows:

$$r_1 + r_2 + ... + r_M = \frac{1}{N} \sum_{t=0}^{N-1} |r_1(t)|^2 + \frac{1}{N} \sum_{t=0}^{N-1} |r_2(t)|^2 + ... + \frac{1}{N} \sum_{t=0}^{N-1} |r_M(t)|^2, \tag{4}$$

$$y_i = \frac{1}{N} \sum_{t=0}^{N-1} |r_i(t)|^2, \tag{5}$$

where $N$ is the total number of samples which is defined as $N = 2T_sW$; here, $T_s$ is the sensing time in ms and $W$ is the PU signal bandwidth in kHz.

Based on Equation (5), the received energy, $y_i$ follows a central Chi-square distribution with $N$ degree of freedom, which is defined as

$$y_i = \begin{cases} X_N^2; & : H_0, \\ X_N^2(N\lambda_i); & : H_1, \end{cases} \tag{6}$$

where $\lambda_i$ is the SNR at the $i$th CU, which is defined as $\lambda_i = \frac{E_s \sum_{t=0}^{N-1} |h_i(t)|^2}{N}$; here, $E_s$ is the signal power of $s(t)$ i.e., $E_s = \sum_{t=0}^{N-1} |s(t)|^2$.

Using the central limit theory (CLT), if $N \geq 200$, then the received signal $y_i$ at the $i$th CU becomes a Gaussian random variable with mean $(x_{0i}, x_{1i})$ and variance $(V_{0i}, V_{1i})$ under both hypotheses $H_0$ and $H_1$, respectively [5], as follows:

$$\begin{cases} x_{0i} = N; & V_{0i} = 2N, \\ x_{1i} = N(1 + \lambda_i); & V_{1i} = 2N(1 + 2\lambda_i). \end{cases} \tag{7}$$

## 5. Proposed Scheme Based on Unscented Kalman Filter Using an Adaptive Fuzzy System

### 5.1. Cooperative Spectrum Sensing (CSS)

The CSS is an efficient solution to enhance the detection performance, in which the CUs collaborate to sense the licensed channel for finding the spectrum holes. In the proposed scheme, we consider two systems, i.e., a linear system and a nonlinear system. We can estimate a linear system based on KF. As a result, we can not detect the spectrum hole properly in a nonlinear system. We proposed UKF based on CSS in a CRN using an adaptive fuzzy system where the UKF estimates a nonlinear system. The diagram of proposed UKF based on CSS in a CRN using an adaptive fuzzy system is shown in Figure 4.

We received local observation energy $y_i$ at the $i$th CU, which is fuzzified by two domains, which are defined as low $\mu_{low}(y_i)$ and high $\mu_{max}(y_i)$. The inference rules are used to make local decisions, $pd_i$ based on fuzzified energy. Then, we makes a global decision, $gd$, based on the defuzzified. When the value of $gd = 1$, then the proposed UKF updates the fuzzification parameters.

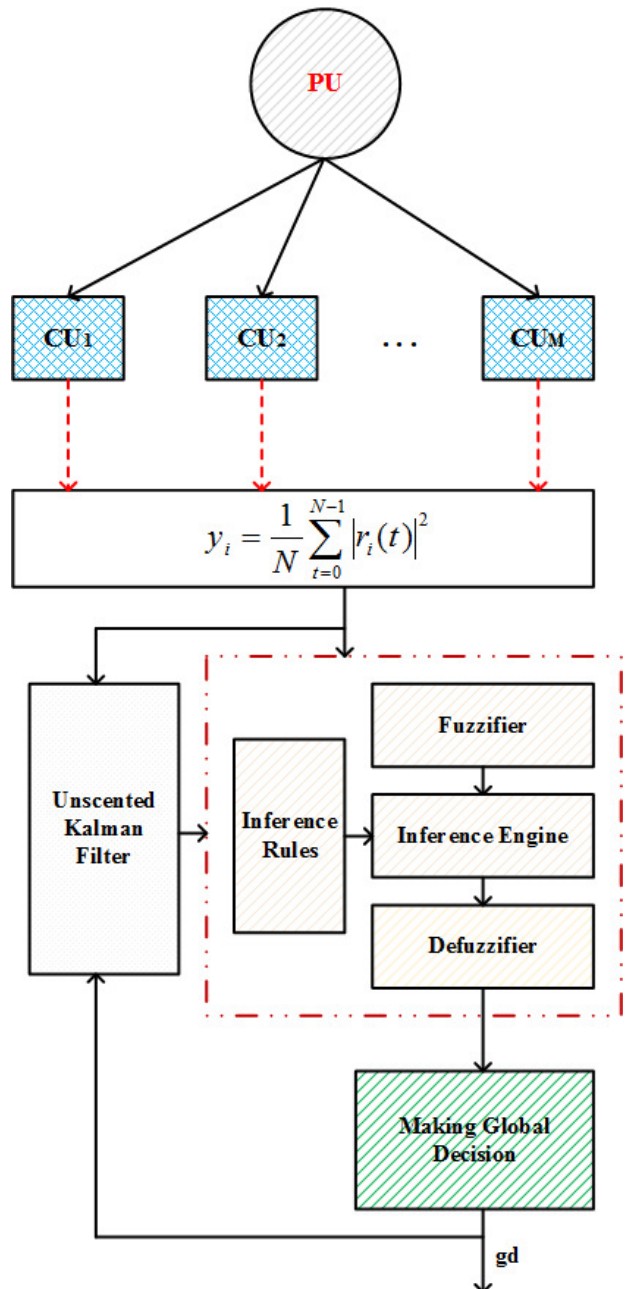

**Figure 4.** The proposed unscented Kalman filter (UKF) based on cooperative spectrum sensing (CSS) in a cognitive radio network (CRN) using an adaptive fuzzy system.

*5.2. Kalman Filter (KF)*

The KF performs based on a recursive algorithm that is used to minimizing mean squared estimation error [48]. In a state space model, it estimates an unknown state variable based on noisy measurement. The KF calculates the posterior estimate from the previous estimate state space. The KF is optimal in a linear system. In an estimation problem, the state space model may have a nonlinear process model and/or a nonlinear measurement model. However, the KF does not work in a nonlinear system. We consider a linear system as follows:

$$y = H \times x, \tag{8}$$

$$E[y] = E[H \times x] = HE[x]. \tag{9}$$

Now, we consider a nonlinear system as follows:

$$y = H(x), \tag{10}$$

$$E[y] = E[H(x)] \neq H[E(x)]. \tag{11}$$

In this case, the KF is only able to estimate in a linear system. However, the KF fails to converge to the true value in a nonlinear system. This problem is mitigated by our proposed UKF scheme.

*5.3. Unscented Kalman Filter (UKF)*

The UKF has been used as the standard technique for performing recursive nonlinear estimation [25,27]. The proposed UKFs are derivative from the EKF which performs to outstanding in a nonlinear system. Now, we consider a nonlinear system as follows:

$$x(k+1) = F(x(k)) + w_n, \tag{12}$$

$$y(k) = H(x(k)) + v_n, \tag{13}$$

where $F(.)$ and $H(.)$ represent nonlinearity function in the process and measurement model, respectively; $w_n$ and $v_n$ represent the process noise and measurement noise, respectively.

The proposed Algorithm 1 based on UKF uses unscented transformation (UT) to capture the propagation of the statistical properties of state estimates through nonlinear functions. In Algorithm 1, initially, we generate a set of state values that are called sigma points. These sigma points express the mean and covariance of the state estimates. This algorithm uses each of the sigma points as an input to the state transition and measurement functions to get a new set of transformed state points. The mean and covariance of the transformed points are then used to obtain state estimates and state estimation error covariance. The main challenge is to detect the spectrum hole properly. Thus, we have applied the observed energy, $y_i$, at the $i$th CU based on the UKF. At the $i$th CU, we initialized an initial estimate $x_0^i = x_{1i}$ and covariance $p$ properly. Then, the filter rotates in several iterations until the filter converges to the true values or approximate true values. In each iteration, the sigma points $x_{jk}$ with $j = 1, 2, \ldots n$ are selected perfectly to express the mean and covariance of the state variable. A state variable of dimension $n$ requires $2n$ sigma points. After choosing sigma points, the filter executes the predicted step in which the sigma points are propagated through the nonlinear process model. The obtained result is used to predict the posterior estimate, $x^-(k)$.

Now, we calculate the posterior covariance, $p^-(k)$ as follows:

$$p^-(k) = E[(x(j, k+1) - x^-(k))(x(j, k+1) - x^-(k))]. \tag{14}$$

The posterior estimate and covariance are appraised as weighted mean and covariance of the resultant. Now, we calculate the effective weights, which are defined as follows:

$$w^0 = \frac{1}{n+s}, \tag{15}$$

$$w_j^m = w_j^c = \frac{1}{2(n+s)} \ for \ j = 1, 2 \ldots n, \tag{16}$$

$$w_{j+n}^m = w_{j+n}^c = \frac{1}{2(n+s)} \ for \ j = n+1 \ldots 2n, \tag{17}$$

where $s$ is a design parameter. Here, $s = 0$; then, general UT reduces to the basic UT. Then, a filter executes the updating step where the posterior estimate and covariance are corrected by the measurements. They are updated based on the difference between predicted observations, $y^-(k)$ and the actual measurements, $y(j, k)$.

### 5.4. Fuzzy Set

In real environments, we cannot explain by conventional logic theory. We just include all those elements with the only partial membership of a set. However, fuzzy set theory accepts those partial memberships and generalizes the conventional logic theory [23]. Basically, a fuzzy set $\overline{A}$ is defined by the membership function of $\mu_{\overline{A}}$. A membership function is a function that defines how each sigma point is mapped to a membership value that is 0 or 1 as follows:

$$\mu_{\overline{A}} : R \to [0,1]. \tag{18}$$

### 5.5. Fuzzy Logic

Fuzzy logic is one kind of special logic that consists of multiple values. It produces degrees of membership and truth. In addition, it performs AND, OR, and NOT like conventional logic operations. Let $p$ be a proposition and $\mu_{\overline{p}^c}$ be a membership function that is defined as follows:

$$\mu_{\overline{p}^c} = 1 - \mu_{\overline{p}}, \tag{19}$$

$$\mu_{\overline{p} \vee \overline{q}} = max(\mu_{\overline{p}}, \mu_{\overline{q}}), \tag{20}$$

$$\mu_{\overline{p} \wedge \overline{q}} = min(\mu_{\overline{p}}, \mu_{\overline{q}}), \tag{21}$$

$$\mu_{\overline{p} \to \overline{q}} = \mu_{\overline{p}^c \vee \overline{q}} = max(\mu_{\overline{p}^c}, \mu_{\overline{q}}). \tag{22}$$

### 5.6. Fuzzifier

Fuzzification is the process which translates a fuzzy set from the crisp input set [47,48]. We have fuzzified each observed energy, $y_i$, into two fuzzy sets, which are defined as low and high. The membership functions are defined as follows:

$$\mu_{low}(y_i) = \begin{cases} 1, & if\, y_i \leq \check{x}_{0i}, \\ e^{-\frac{(y_i - \check{x}_{0i})^2}{2V_{0i}}}, & otherwise, \end{cases} \tag{23}$$

$$\mu_{high}(y_i) = \begin{cases} 1, & if\, y_i \geq \check{x}_{1i}, \\ e^{-\frac{(y_i - \check{x}_{1i})^2}{2V_{1i}}}, & otherwise. \end{cases} \tag{24}$$

### 5.7. Fuzzy Inference Rules

The inference rules obtain information about the presence of PU signal based on fuzzified energy. Let $pd_i$ denote a local decision that reflects the presence possibility of the PU signal based on the observation at the $i$th CU. Now, we can define the fuzzy inference rule as follows:

- Rule 1: If ($y_i$ is Low), then ($pd_i = pd_{min}$).
- Rule 2: If ($y_i$ is High), then ($pd_i = pd_{max}$).

### 5.8. Defuzzification

Defuzzification is a process of transposing fuzzy output to crisp output. We have used the weighted average method for defuzzification. This method is very suitable for fuzzy sets with symmetrical output membership functions [23]. We have fuzzy sets $pd_{min}$, $pd_{max}$ with fuzzy weighted $\mu_{low}(y_i)$, $\mu_{high}(y_i)$. Then, a local decision can be defined using the weighted average method as follows:

$$pd_i = \frac{pd_{min}\mu_{low}(y_i) + pd_{max}\mu_{high}(y_i)}{\mu_{low}(y_i) + \mu_{high}(y_i)}. \tag{25}$$

We can also simplify a local decision, $pd_i$, which is taking the values in symmetric domain $[-1, 1]$. We can define $pd_{min} = -1$ and $pd_{max} = 1$; here, $pd_i = -1$ is the absence of PU signal and $pd_i = 1$ is the presence of PU signal. Then, we can calculate a local decision, $pd_i$ as follows:

$$pd_i = \frac{-\mu_{low}(y_i) + \mu_{high}(y_i)}{\mu_{low}(y_i) + \mu_{high}(y_i)}. \tag{26}$$

*5.9. Global Decision*

In a CRN, the $i$th CU obtained a local decision, $pd_i$, and then forwards their decisions to the corresponding FC that makes a global decision using the defuzzification procedure. Then, at an FC, we can calculate a global decision, $gd$, as follows:

$$gd_{d|f} = \sum_{i=1}^{M} pd_i. \tag{27}$$

Moreover, a global decision, $gd$, is also defined under hypotheses as follows:

$$gd_{d|f} = \begin{cases} 1, if \sum_{i=1}^{M} pd_i > \tau_{th}; & : H_1, \\ 0, otherwise; & : H_0, \end{cases} \tag{28}$$

where $\tau_{th}$ is a global decision threshold at the FC.

In addition, we can calculate the global probability of error of the proposed scheme as follows [41]:

$$p_E = \alpha gd_f + (1 - \alpha)(1 - gd_d), \tag{29}$$

where $\alpha$ is the probability of the absence of the PU and $(1 - \alpha)$ is the probability of the presence of the PU.

## 6. Simulation Results and Discussion

In this section, we evaluate the performance of the proposed spectrum sensing scheme via Matlab (version 7.2, The MathWorks, Inc., Natick, MA, USA) in which Monte-Carlo simulations were carried out under the following conditions in Table 2.

**Table 2.** Simulation parameters.

| Parameters | Value |
| --- | --- |
| The number of samples, $N$ | 300 |
| The number of iteration, $L$ | 5000 |
| The sensing time, $T_s$ | 1 ms |
| The time slot length, $T$ | 10 ms |
| The channel bandwidth, $W$ | 300 kHz |
| The number of CUs, $M$ | [5, 10] |
| The primary user signal, $s(t)$ | BPSK |
| The channel noise in CU, $\omega(t)$ | AWGN |
| The minimum SNR, $\lambda_{min}$ | $-30$ dB |
| The maximum SNR, $\lambda_{max}$ | 20 dB |
| The channels | AWGN fading |
| The global decision threshold, $\tau_{th}$ | $[-5, 5]$ |
| The probability of the absence of the PU, $\alpha$ | 0.5 |
| The probability of the presence of the PU, $(1 - \alpha)$ | 0.5 |

Firstly, we evaluate the nonlinear state space model based on UKF. The state space estimation of nonlinear case and the error covariance are shown in Figure 5. In Figure 5, we have shown actual measurement at the $i$th *CU* and estimated value based on UKF under the alternative hypothesis $H_1$.

It is clear that the actual measurement and the estimated value using UKF are approximately the same. In Figure 6, we show the mean squared error minimization (MSEM) based on UKF at the $i$th $CU$ under the alternative hypothesis, $H_1$. In this case, the mean square estimation error is decreasing when increasing the number of iterations.

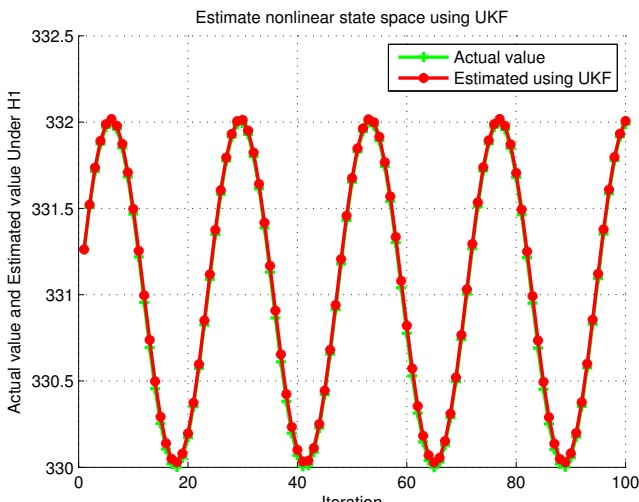

**Figure 5.** Converge estimated value to the actual value in a nonlinear system based on unscented Kalman filter (UKF) under the alternative hypothesis, $H_1$ at the $i$th $CU$ with $n = 2$.

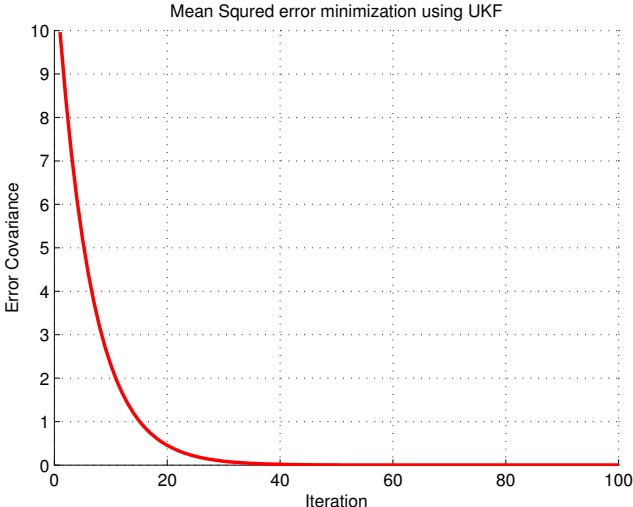

**Figure 6.** The mean squared error minimization (MSEM) in a nonlinear system based on unscented Kalman filter (UKF) under the alternative hypothesis, $H_1$ at the 1st $CU_1$ with $n = 2$.

Secondly, in Figure 7, we present the receiver operating characteristics (ROC) curves for different numbers of CUs, ($M = 5$) with different SNRs ($[-8, -11, -14, -17, -20]$). In the condition of these SNRs, Figure 7 shows the detection performance of the proposed scheme ($gd_d = 0.69$) under an AWGN channel when the probability of false alarm, ($gd_f = 0.1$), is always better than both the conventional GEC scheme ($gd_d = 0.53$) and the conventional KF scheme ($gd_d = 0.60$). Moreover, we show in Figure 8 where we can plot the global probability of error, $p_E$, which is a function of the probability of detection, $gd_d$. We can clearly see that the proposed scheme achieved the minimum global probability of error, ($p_E = 0.1$) at the probability of detection, ($gd_d = 0.8$), compared to both the conventional EGC scheme ($p_E = 0.24$) and the conventional KF scheme ($p_E = 0.22$).

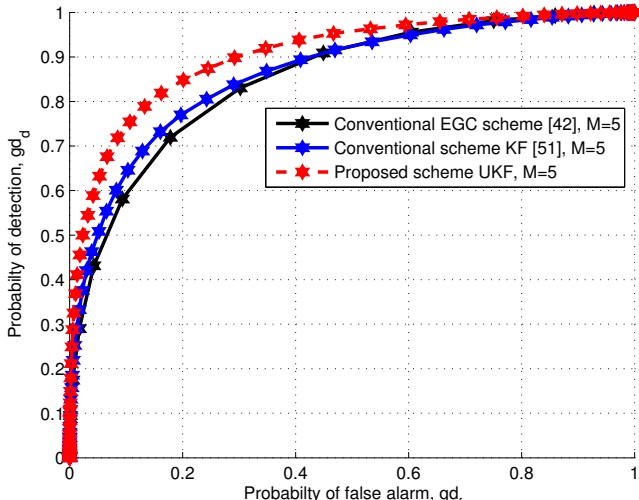

**Figure 7.** Detection gain comparison between an equal gain combining (EGC), Kalman filter (KL) and unscented Kalman filter (UKF) with different numbers of CUs, i.e., $M = 5$ and signal to noise ratio (SNR) = $[-8, -11, -14, -17, -20]$.

In Figure 9, we present the ROC curves for different numbers of CUs, ($M = 10$) with different SNRs ($[-8, -11, -17, -20, -8, -11, -14, -17, -20]$). In the condition of these SNRs as shown in Figure 9, the detection performance of the proposed scheme ($gd_d = 0.91$) under an AWGN channel when the probability of false alarm, ($gd_f = 0.1$) is always better than both the conventional EGC scheme ($gd_d = 0.77$) and the conventional KF scheme ($gd_d = 0.81$). Moreover, we show in Figure 10 where we can plot the global probability of error, $p_E$, which is a function of the probability of detection, $gd_d$. We can clearly see that the proposed scheme achieved the lowest global probability of error, ($p_E < 0.11$), at the probability of detection, ($gd_d = 0.8$), compared with both the conventional EGC scheme ($p_E = 0.16$) and the conventional KF scheme ($p_E = 0.14$).

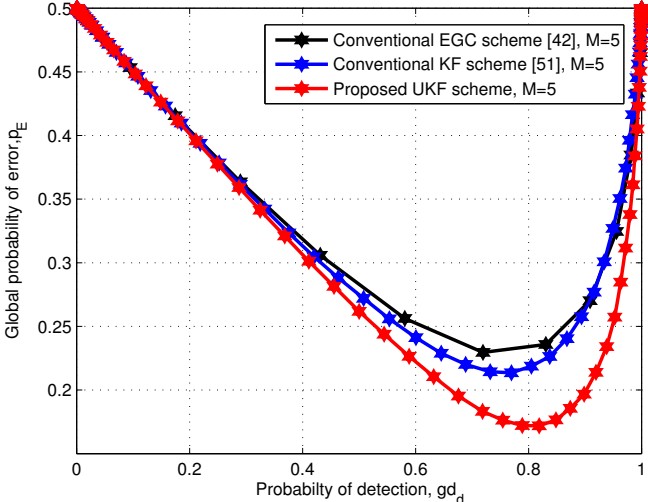

**Figure 8.** Probability of error comparison between an equal gain combining (EGC), Kalman filter (KL) and unscented Kalman filter (UKF) with different numbers of CUs, i.e., $M = 5$ and signal to noise ratio (SNR) = $[-8, -11, -14, -17, -20]$.

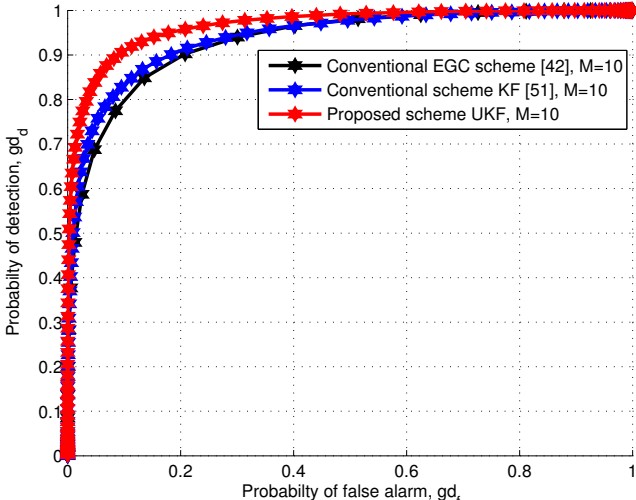

**Figure 9.** Detection gain comparison between an equal gain combining (EGC), Kalman filter (KL) and unscented Kalman filter (UKF) with different numbers of CUs, i.e., $M = 10$ and signal to noise ratio (SNR) = $[-8, -11, -14, -17, -20, -8, -11, -14, -17, -20]$.

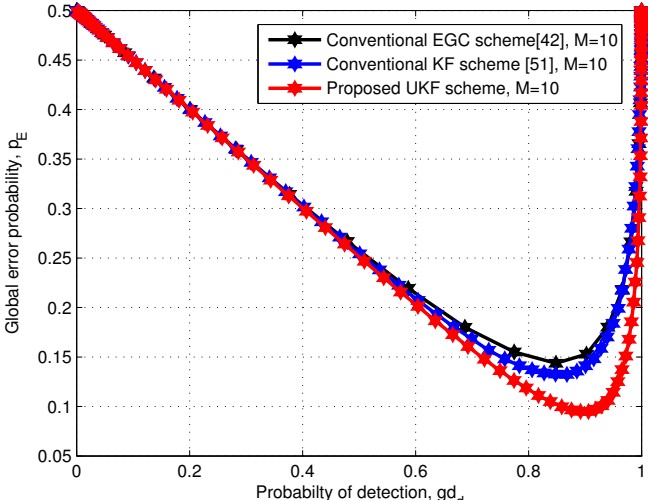

**Figure 10.** Probability of error comparison between an equal gain combining (EGC), Kalman filter (KL) and unscented Kalman filter (UKF) with different numbers of CUs, i.e., $M = 10$ and signal to noise ratio (SNR) = $[-8, -11, -14, -17, -20, -8, -11, -14, -17, -20]$.

From both Figures 8 and 10, a higher global probability of error, $p_E$, means that lower spectrum efficiency for both the PUs and CUs, and vice versa, is true. Therefore, the lowest global probability of error, $p_E$, of the proposed scheme enhances detection gain compared to both the EGC scheme and KF scheme.

Thirdly, in Figure 11, the proposed scheme can obtain better detection gain compared to the KF scheme under different threshold values. It is easy to understand that the detection gain, $gd_d$, remains constant i.e., approximately 1 under the threshold value, from $-5$ to $-1.5$, i.e., $\tau_{th} = [-5, -1.5]$, while the probability of detection, $gd_d$, decreases under the threshold value, from $-1.5$ to 4 i.e., $\tau_{th} = [-1.5, 4]$.

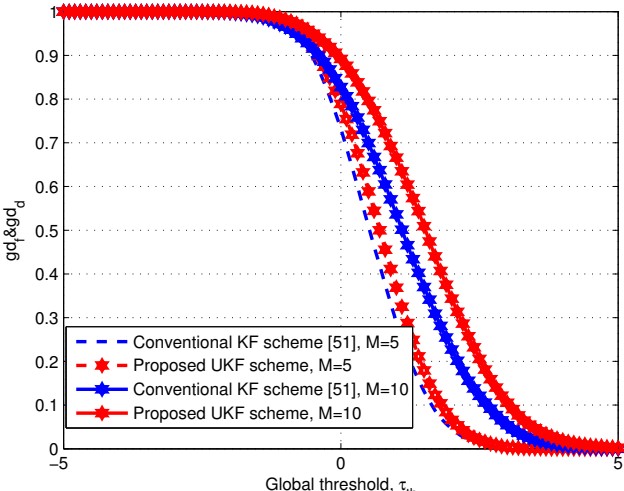

**Figure 11.** Performance comparison between Kalman filter (KL) and unscented Kalman filter (UKF) with different numbers of CUs.

Finally, in Figure 12, the proposed UKF scheme outperforms the conventional KF scheme under a non-fading channel i.e., AWGN channel. Similarly, under both Rayleigh fading channel and Log-normal shadow fading channel conditions, the proposed UKF scheme achieved a better detection gain compared with the conventional KF scheme.

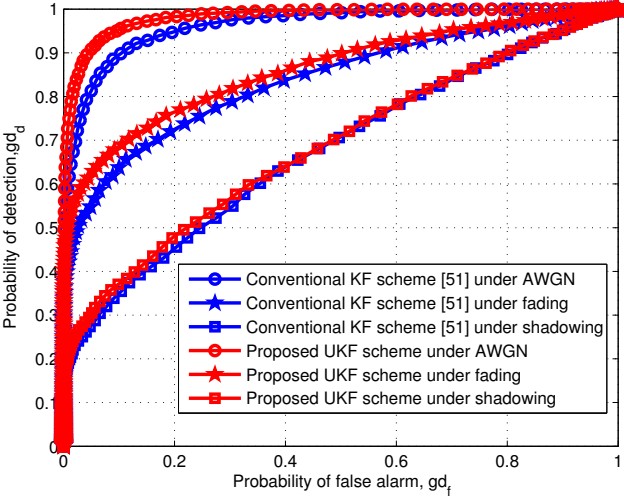

**Figure 12.** Detection gain comparison between Kalman filter (KL) and unscented Kalman filter (UKF) under different fading channels.

## 7. Conclusions and Future Work

In this paper, the proposed UKF scheme can achieve better sensing performance and the lowest global probability of error compared to both the conventional EGC scheme and conventional KF scheme. In the detection gain, the detection probability of the proposed UKF scheme is 18.18% and 12.34% over the conventional EGC scheme and the conventional KF scheme, respectively. In addition, the global probability of error in the proposed UKF scheme is 31% and 21% less than the conventional EGC scheme and conventional KF scheme, respectively. Moreover, the detection probability of the proposed scheme is more outstanding than the conventional KF scheme under the global threshold value, i.e., $\tau_{th} = [-1.5, 4]$. Therefore, we conclude that our proposed UKF scheme will be more

affordable and applicable in CRNs due to alleviating the spectrum deficiency problem and minimizing the global probability of error.

For future work, we will analyze the minimize time accuracy of the proposed UKF scheme. In addition, we will analyze the computational complexity of the proposed UKF scheme in comparison with existing techniques.

**Author Contributions:** The idea was formulated by M.R.A.; the CRN system was designed, experiments and performed the experiments by M.R.A., M.A.H. and M.S.M.; M.M.R. and M.S.M. are both supervised the research; M.R.A., M.A.H., M.K.I., K.M.A. and M.S.M. wrote the paper; M.M.R., B.C.S. and M.S.M. were analytically reviewed and revised the manuscript.

**Funding:** This work was supported by the Department of Information and Communication Engineering (ICE), Islamic University, Kushtia 7003, Bangladesh.

**Conflicts of Interest:** The authors declare no conflict of interest.

## Abbreviations

The following abbreviations are used in this manuscript:

| | |
|---|---|
| UKF | Unscented Kalman Filter |
| CSS | Cooperative Spectrum Sensing |
| CRN | Cognitive Radio Network |
| PU | Primary User |
| FC | Fusion Center |
| EGC | Equal Gain Combining |
| KF | Kalman Filter |
| WSN | Wireless Sensor Network |
| FCC | Federal Communication Commission |
| CR | Cognitive Radio |
| CRN | Cognitive Radio Network |
| CU | Cognitive User |
| ED | Energy Detection |
| SNR | Signal-to-Noise Ratio |
| EKF | Extended Kalman Filter |
| AWGN | Additive White Gaussian Noise |
| PN | Primary Network |
| TDMA | Time Division Multiplexig Access |
| ADC | Analog-to-Digital converter |
| CLT | Central Limit Theory |
| MSEM | Mean Squared Error Minimization |
| ROC | Receiver Operating Characteristics |

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
