# Peer review of "Unscented Kalman Filter Based on Spectrum Sensing in a Cognitive Radio Network Using an Adaptive Fuzzy System"

_2504-2289, doi:10.3390/bdcc2040039_

Round 1

Reviewer 1 Report

In this paper, the authors proposed the use of unscented kalman filter (UKF) based cooperative spectrum sensing (CSS) in a cognitive radio network (CRN) using adaptive fuzzy system to improve the detection of the primary user signals. This topic is very well developed in the literature and there are plenty of works done on spectrum sensing. This paper in its current form is not suitable to be published as a journal paper because the whole idea of the paper is not clear. It is also not clear how UKF along with fuzzy system helps improve the PU detection. English is also another issue in this paper. The other comments are listed as follows:

- It is not correct to mention Algorithm 1 in the abstract. Abstract is just a summary of the paper and it should not refer the reader to any specific section of the paper.

-The paper needs extensive English editing. There are a lot of structural and  grammatical mistakes.

-The authors used (i), (ii), … to refer to the sensing methods instead of writing the name of the methods.

-In Line 54, what is the reference for claiming lower SNR of -28dB?

-All the statements in lines 57 to 61 need proper references. There is also a paradox in the statements in these lines about SNR estimation in the CU.

-The authors should mention what type of spectrum sensing they have used.

-In the introduction, the authors need to briefly clarify how UKF works better than KF and EKF.

-The authors used “we can propose…..” in several places which make the contribution of the paper vague.

-Sections 3 and 4 are redundant as there are a large number of papers in the literature that discuss the energy detection.

-Figures 1 to 4 are blurry.

-Future work is not well written.

Author Response

Dear Reviewer,

Please could you find the attached file.

Reviewer 2 Report

Proposal to transmit measured energy to FC is very costly, and increases control traffic  -"each CU measures the energy of the received signal in the band of interest and then transmits its observation to the FC without any extra information"  

Chapter 2 - related works are described, but without or with very limited analysis and comparison

line 126 - please explain assumption that the signal is BPSK

The simulation scenario description is too simplified. Presented simulation results doesn't take into account attenuations between nodes, fading and shadowing effects, being a reason of cooperate sensing. 

Editorial comments:

line 20 wireless applications that are available for fulfilling the users demand

line 24 mitigates -> mitigate … all sentence should be reviewed

line 27 in which CU to access the licensed spectrum .sentence should be reviewed

etc.

general comment: paper requires professional proofreading.

Author Response

Dear Reviewer,

Please could you find the attached file.

Best regards

Md Sipon Miah

Reviewer 3 Report

In this work, an unscented Kalman Filter based cooperative spectrum sensing technique that uses an adaptive Fuzzy system is presented for cognitive radio. The unscented Kalman filter presents better performance when estimation under the assumption of non-linearity is perfomed compared to the case when the traditional Kalman filter is used. The results shown in the simulations’ section verify the previous gains. I have though the following comments that need to be addressed.

1.      The literature of spectrum sensing and collaborative spectrum sensing is quite mature. To that end, I found the list of references quite dated. I suggest, to improve the corresponding literature survey with more state of the art works. Please find below some recommendations.

a.      C. G. Tsinos and K. Berberidis, "Decentralized Adaptive Eigenvalue-Based Spectrum Sensing for Multiantenna Cognitive Radio Systems," in IEEE Transactions on Wireless Communications, vol. 14, no. 3, pp. 1703-1715, March 2015.

b.      C. G. Tsinos and K. Berberidis, "Adaptive Eigenvalue-Based Spectrum Sensing for multi-antenna cognitive radio systems," 2013 IEEE International Conference on Acoustics, Speech and Signal Processing, Vancouver, BC, 2013, pp. 4454-4458.

c.      C. Politis, S. Maleki, C. G. Tsinos, K. P. Liolis, S. Chatzinotas and B. Ottersten, "Simultaneous Sensing and Transmission for Cognitive Radios With Imperfect Signal Cancellation," in IEEE Transactions on Wireless Communications, vol. 16, no. 9, pp. 5599-5615, Sept. 2017.

2.      Computational complexity analysis of the proposed techniques is completely missing. A comparison to the complexity of existing techniques should be also added.

3.      In the simulations, results should be added that show the performance of the depicted techniques with respect the number of nodes. At the present version, results only for 5 and 10 nodes are shown.

4.      The manuscript is very poorly written. There are lots of grammar and syntax errors, please revise accordingly.

Author Response

(The authors gave the same response as above.)

Reviewer 4 Report

The manuscript presents two main issues: technical content and novelty.

As regards to the former issue, given the wide literature about spectrum sensing in cognitive radio scenarios, the authors fail in describing the novelty of the work with respect to the literature. Indeed, the reader's impression is mostly an heuristic work based on UKF and fuzzy logic.

As regards to the latter issue, which indeed is related to the first one, the references are mainly either un-relevant or outdated (with ref [6] not even completed), and the authors should significantly improve them, by discussing recent relevant works such as [A-D] and the references therein

[A] http://dx.doi.org/10.1109/TWC.2011.081011.102164

[B] http://dx.doi.org/10.1109/TWC.2013.031813.121112

[C] http://dx.doi.org/10.1109/JSAC.2013.131102

[D] http://dx.doi.org/10.1109/JSAC.2012.120208

Author Response

Dear Reviewer,

Please could find the attached file.....

Round 2

Reviewer 1 Report

The comments have addressed to some extent. However, there are still several problems: 

1- The paper needs extensive English editing. It still contains numerous grammatical and syntax errors. In some places, the English is even worse than in the first version of the paper.

2- Regarding the references, the authors added five references from 2008, 2010, 2004, 2010, and only one from 2016.

3- One other problem in the references is the addition of  4 references from Tsinos et. al. and 4 references from Cacciapuoti et. al. Authors are not supposed to cite many times the same groups as there are many other research papers that are worth citing. 

Author Response

Dear Reviewer Sir,

Please find attached file .....

Best regards

Sipon

Reviewer 3 Report

The authors have addressed all of the comments successfully and thus, I recommend the manuscript acceptance.

Author Response

Dear Reviewer,

Please find attached file..

Best regards

Md Sipon Miah

Reviewer 4 Report

The authors addressed all the raised comments.

Author Response

Dear Reviewer,

Please find attached file.

Best regards

Md Sipon Miah

Round 3

Reviewer 1 Report

Comments not completely addressed. Paper has major flaws and is poorly written.

Author Response

Dear Reviewer,

Kind regards

Md Sipon Miah
